# Bounding Spill-over Effect under Structural Uncertainty

## Abstract

Causal inference on graphs has attracted increasing attention due to possible interactions among units. One main challenge is the spill-over effects, i.e., the influence of treatments on neighboring nodes on the target outcome. However, the observed graphs may suffer from inconsistency between local network structure and the interference mechanism, which invalidates the identifiability guaranty of existing spill-over estimators. To address the challenge, we propose learning to bound spill-over effects under local structural uncertainty, which aims to obtain optimization-based bounds for uncertain nodes over a learned feasible set of probably consistent ego-graphs. Specifically, we start by introducing a structure proposal network that maps the ego-graph of uncertain nodes to probably consistent candidate graphs, where a spill-over effect estimator is introduced to explore the upper and lower limits of the spill-over effect by traversing the feasible ego-graph space defined by the generator. The generated ego-graphs are constrained to preserve the incomplete information (**closeness**) and be indistinguishable from the consistent ego-graphs sampled from stable nodes (**consistency**), whereas the spill-over estimator is constrained to be compatible with the observed outcomes on the network (**faithfulness**). We formulate the above objectives as a constrained, bi-level adversarial learning framework, where an efficient and stable EM-based objective is proposed to solve the optimization problem. Experiments on both simulated and semi-simulated datasets show the effectiveness of the proposed method.

## 1   Introduction

Causal inference studies the cause-and-effect relationship between a treatment and its outcome (Yao et al., 2021). Traditional causal inference generally assumes non-interference among the units, i.e., the treatment assigned to one unit cannot causally influence the outcomes of other units (Cox, 1958). Although the assumption holds in certain cases, in many real-world observational studies, interactions form among the units, where treating one unit could have *spill-over* effects on other units through interactions (Hudgens & Halloran, 2008). For example, an advertising campaign targeting on specific users on a social media can inadvertently influence their connected peers if connected users communicate about the Ads in real life (Sahni, 2016). Hence, it is important to model the spill-over effects, as it facilitates more accurate treatment decisions by comprehensively considering the overall interactions for the entire population on the graph.

Generally, existing causal inference methods that model spill-over effects on graphs either adopted statistical adjustment, e.g., incorporating the treatments of neighboring nodes (Toulis & Kao, 2013) or the proportion of treated neighbors (Tchetgen & VanderWeele, 2012) as extra covariates to adjust for the interference, or used representation learning models such as graph neural networks (GNNs) (Wu et al., 2020) to summarize the interference patterns in the outcome prediction model (Ma et al., 2022; Ma & Tresp, 2021). Here, an implicit assumption made by these methods is *structural consistency*, i.e., the observed graph structure is consistent with the true interference mechanism (Hamilton et al., 2017). However, real-world graphs often suffer from *local structural uncertainty*, i.e., the observed neighborhood can be untrustworthy for certain non-stable nodes in the graph (Chen et al., 2020; Fatemi et al., 2021). For example, the observed connections for new users in a social network may not faithfully reflect how they communicate in real life, as these users may not get the time to add all the real-world connections. Consequently, the local structure around these nodes could mis-specify the true interference structure, which precludes the existing methods from unbiasedly estimating the spill-over effects.

Recently, preliminary explorations have been conducted on estimating the spill-over effects under structural uncertainty. For example, if multiple possible interference graphs exist, denoising models can be used to recover the true interference structure (Li et al., 2021; Weinstein & Nevo, 2023). If only a single graph can be observed, (Vazquez-Bare, 2023; Yu et al., 2022) show that spill-over effect can be estimated only if treatments are randomly assigned (Athey & Imbens, 2017). However, in many cases, randomized experiments are too expensive to conduct on graphs. In addition, these works mainly focus on the *average* spill-over effect, i.e., the expected difference of spill-over effects between treated and control nodes. This provides less information on the individualization of treatments. Finally, existing methods only provide point estimation of the spill-over effect even if uncertainty exists. For high-stake causal inference tasks where interference exists, e.g., treatment effect estimation on military (Papadogeorgou et al., 2022) or economic graph (Varian, 2016), it is important to have conservative/optimistic optimization-based bounds to inform better decisions.

To address the above challenges, we propose learning to bound spill-over effects under local structural uncertainty, which aims to provide optimization-based bounds for uncertain nodes over a learned feasible set of probably consistent ego-graphs. We first introduce a novel structural proposal network (SPN) that generates probably consistent ego-graph candidates for the uncertain nodes from their observed uncertain local structure. In addition, a spill-over effect estimator is introduced accordingly to explore the upper and lower limits of the spill-over effect for each uncertain node based on the probably consistent ego-graph distribution specified by the SPN. The exploration space is constrained such that the generated probably consistent ego-graph candidates preserve the useful information in the observed local structure (**closeness**) while being indistinguishable from the consistent ego-graphs distribution specified by the stable nodes (**consistency**), whereas the spill-over effect estimator remains compatible with the observed outcomes on the graph (**faithfulness**). Furthermore, based on an *Unconfoundeness under the Structural Recovery assumption*, we connect the faithfulness constraint with the spill-over bound exploration objective and formulate the above task as a bi-level, constrained optimization problem, which we demonstrate can be effectively solved with an EM-like objective. Extensive experiments on both semi-simulated and real-world graphs demonstrate its significantly enhanced robustness to local structural uncertainty[1].

## 2 Preliminaries

### 2.1 Problem Formulation

Consider an observational study with $N$ units $\mathcal{N} = \{i\}_{i=1}^{N}$[2], where $\mathbf{t} \in \{0,1\}^N$ denotes the observed treatments, $\mathbf{y} \in \mathbb{R}^N$ denotes the observed outcomes, $\mathbf{X} \in \mathbb{R}^{N \times K_X}$ denotes the pre-treatment covariates, and $T$, $X$, $Y$ denote the corresponding random variables, respectively. Our aim is to estimate the causal relationship between $T$ and $Y$, given that *treatment interference* exists among the units in $\mathcal{N}$ due to inter-unit interactions. The observed interactions among the units form a graph $G = (\mathcal{N}, \mathbf{A})$, where $\mathbf{A}$ is the adjacency matrix and $A_{i,j} = 1$ denotes that interaction between node $i$ and node $j$ is observed and $A_{i,j} = 0$ otherwise. Throughout this paper, we assume **locality** of the spill-over effect, i.e., treatment $T_i$ can only influence up to $K$-hop interactions of node $i$. The *observed* local structure centered at node $i$ can be represented as an ego-graph $G_{i,K}$, i.e., the sub-graph of $G$ that contains the node $i$ and all its $K$-hop neighbors. For notational simplicity, we omit $K$ whenever it is clear from the context. To connect the *observed* local ego-graph structure with the *true* interference mechanism, we define the concept of "consistency" as:

**Definition 1. (Consistency)** *An observed ego-graph $G_i$ is said to be consistent with the interference mechanism for node $i$ if and only if all the nodes within $K$-hop interactions with node $i$ are included in $G_i$ with interactions recorded as edges in its adjacency matrix.*

We denote the set of uncertain nodes whose ego-graphs are not consistent as $\mathcal{N}_u$. For an arbitrary node $i$, we denote the *true* consistent ego-graph as $\hat{G}_i$, which may be unobserved and does NOT necessarily share the same nodes as the observed ego-graph $G_i$. Our purpose is to provide bounds for the spill-over effect of the uncertain nodes despite the local structural uncertainty, such that conservative/optimistic estimates of the treatment effect can be obtained to inform better decisions.

---

[1]The code is available in `https://anonymous.4open.science/r/SpillBound-6B41`.
[2]We summarize the notations used in this section in Table 2 in the Appendix.

## 2.2 Ideal Case with Consistent Local Structure

### 2.2.1 Potential Outcome under Networked Interference

In this section, we provide a theoretical analysis of the spill-over effect *if all the ego-graphs in $G$ are consistent*, which is a fragile assumption on which most existing spill-over effects rely. We reason with the treatment effects using Rubin's causal model (RCM) (Rubin, 1974). When no interference exists, RCM defines potential outcome (PO) $Y_i^t$ as the outcome for unit $i$ if treatment $t \in \{0, 1\}$ had been imposed on the unit $i$. With $Y_i^t$, individual treatment effect for unit $i$ can be defined as $\tau_i = Y_i^1 - Y_i^0$. However, when interference exists, the outcome of unit $i$ can also be causally determined by the treatments received by other units within $K$-hop consistent ego-graph of $i$, i.e., $\hat{G}_i$. Therefore, we first extend the definition of PO to allow for the interference:

**Definition 2.** *We use potential outcome $Y_i^{t_i, \mathbf{t}_{-i}}$ to denote the outcome for unit $i$ if treatment $t_i$ had been received by unit $i$ itself and treatments $\mathbf{t}_{-i}$ had been received by other units.*

Specifically, based on the locality assumption, $Y_i^{t_i, \mathbf{t}_{-i}}$ can be represented by a response function:

$$\Phi(t_i, \mathbf{t}_{-i}^{\hat{G}_i}, \mathbf{x}_i, \mathbf{X}_{-i}^{\hat{G}_i}, \hat{G}_i),$$

where $\{\mathbf{t}, \mathbf{X}\}_{-i}^{\hat{G}_i}$ selects rows in $\{\mathbf{t}, \mathbf{X}\}$ by nodes in the consistent ego-graph $\hat{G}_i$. The response function $\Phi$ encapsulates the interventional response of the outcome $Y_i$ if self-treatment $t_i$ and neighbor-treatments $\mathbf{t}_{-i}^{\hat{G}_i}$ had been imposed on the nodes in the consistent ego-graph $\hat{G}_i$. To avoid symbol overload, the superscript $\hat{G}_i$ that confines the variables to nodes in $\hat{G}_i$ will be omitted if no ambiguity exists.

### 2.2.2 Direct and Spill-over Effects

In RCM, treatment effects are defined by comparing the POs under different treatments. When interference exists, two types of treatment effects entangle that simultaneously determine the observed outcome $Y_i$. We first define the (individual) direct treatment effect for a specific node $i$ as follows:

**Definition 3.** *The individual direct effect $\tau^d$ of the treatment $t_i$ on the outcome $Y_i$ is defined as the difference of potential outcomes $Y_i^{t_i, \mathbf{t}_{-i}}$ corresponding to $t = 1$ and $0$ as follows:*

$$\tau^d(\mathbf{t}_{-i}) = \mathbb{E}\left[Y_i^{1, \mathbf{t}_{-i}} - Y_i^{0, \mathbf{t}_{-i}} \Big| \mathbf{x}_i, \mathbf{X}_{-i}, \hat{G}_i\right] = \Phi(1, \mathbf{t}_{-i}, \mathbf{x}_i, \mathbf{X}_{-i}, \hat{G}_i) - \Phi(0, \mathbf{t}_{-i}, \mathbf{x}_i, \mathbf{X}_{-i}, \hat{G}_i). \tag{1}$$

Here, we clarify that we follow previous works to define the individual treatment effect (ITE) as the conditional average treatment effect (CATE) on the pre-treatment variables that describe the unit characteristics (Guo et al., 2020; Ma & Tresp, 2021; Ma et al., 2022). Furthermore, the individual spill-over effect can be defined as follows:

**Definition 4.** *(Individual Spill-Over Effect) The individual spill-over effect $\tau^s$ of node $i$ under the self-treatment $t_i$ is defined as the difference of potential outcomes $Y_i^{t_i, \mathbf{t}_{-i}}$ as:*

$$\tau^s(t_i, \mathbf{t}_{-i}) = \mathbb{E}\left[Y_i^{t_i, \mathbf{t}_{-i}} - Y_i^{t_i, \mathbf{0}} \Big| \mathbf{x}_i, \mathbf{X}_{-i}, \hat{G}_i\right] = \Phi(t_i, \mathbf{t}_{-i}, \mathbf{x}_i, \mathbf{X}_{-i}, \hat{G}_i) - \Phi(t_i, \mathbf{0}, \mathbf{x}_i, \mathbf{X}_{-i}, \hat{G}_i). \tag{2}$$

### 2.2.3 Identification and Estimation

Eqs. (1) and (2) cannot be directly used to estimate $\tau^s$ and $\tau^d$, as only the PO that corresponds to the assigned treatment $\mathbf{t}$, i.e., $Y_i^{t_i, \mathbf{t}_{-i}}$, can be observed for each unit. However, it can be proved that if all confounders, i.e., variables that simultaneously determine the treatment $T$ and the outcome $Y$, are included in the covariates $X$ and graph structure $\hat{G}_i$, which can be formulated as the *Unconfoundedness under Structural Recovery (S.R.)* assumption as follows:

**Assumption 1.** *(Unconfoundedness under S.R.)* $Y^{T_i, T_{-i}} \perp T_i, T_{-i} | X_i, X_{-i}, G_i = \hat{G}_i$,

$\tau^s$ and $\tau^d$ can be estimated from the observational graph data as follows:

**Theorem 2.1.** *(Identification under S.R.)* *If Assumption 1 holds, we have the following identification:*

$$
\begin{aligned}
\tau^d(\mathbf{t}_{-i}) &= \mathbb{E}[Y|1, \mathbf{t}_{-i}, \mathbf{x}_i, \mathbf{X}_{-i}, \hat{G}_i] - \mathbb{E}[Y|0, \mathbf{t}_{-i}, \mathbf{x}_i, \mathbf{X}_{-i}, \hat{G}_i], \\
\tau^s(t_i, \mathbf{t}_{-i}) &= \mathbb{E}[Y|t_i, \mathbf{t}_{-i}, \mathbf{x}_i, \mathbf{X}_{-i}, \hat{G}_i] - \mathbb{E}[Y|t_i, \mathbf{0}, \mathbf{x}_i, \mathbf{X}_{-i}, \hat{G}_i].
\end{aligned} \tag{3}
$$

Proof of Theorem 2.1 is provided in the Appendix. Since the RHS of Eq. (3) does not contain any POs, the expectations can be estimated via models such as graph neural networks (GNNs) (Wu et al., 2020). However, due to local structural uncertainty, the observed ego-graph $G_i$ could mis-specify the true interference mechanism $\hat{G}_i$, where Assumption 1 is violated that prevents identification. To address the above challenge, in the following section, we introduce a new framework that learns to bound the spill-over effect under structural uncertainty, such that conservative/optimistic estimates of the spill-over effects can be obtained for the uncertain nodes with mis-specified interference structures to inform better decisions.

## 3 Methodology

### 3.1 Structural Proposal Network

To bound the spill-over effects under local structural uncertainty, we start by introducing a structural proposal network (SPN) that generates candidate ego-graphs $G_i^c$ for the uncertain nodes in $G$, $i \in \mathcal{N}_u$ that are *probably* consistent with the true interference mechanism given the observed graph data. Here, we note that due to local structural uncertainty, consistent ego-graphs $\hat{G}_i$ are not available for the nodes in $\mathcal{N}_u$, so we use the term "probably consistent" to denote the ego-graphs of the uncertain nodes that satisfy certain constraints derivable from the observational graph data (which will be introduced in subsequent parts). The probably consistent ego-graphs form the candidate space from which the spill-over effect estimator explores and finds the bounds of the spill-over effects.

To capture the local structural uncertainty for the nodes in $\mathcal{N}_u$, we parameterize the SPN as a variational *ego*-graph auto-encoder (VEGAE), where a *probabilistic* encoder $Q(Z|G_i, O)$ is introduced to map the uncertain ego-graph $G_i$ centered at node $i$ and other observed covariates $O = \{G_i^a, T, X, Y\}$ (with $G_i^a$ being the ambient graph $G/G_i$) to the latent space $Z$, from which the decoder $P(G_i^c|Z, O)$ generates the candidate ego-graph $G_i^c$ that is consistent with the true interference mechanism. Here, $G_i$ and $G_i^c$ are allowed to have different node sets. Ideally, if exemplar probably consistent ego-graphs $G_i^c = \hat{G}_i$ can be obtained, VEGAE can be directly optimized with $\hat{G}_i$ based on the evidence lower bound (ELBO) (Kipf & Welling, 2016) as follows:

$$
\ln P(\hat{G}_i|G_i, O) \geq \mathbb{E}_Q[P(\hat{G}_i|Z, O)] + \mathbb{KL}[Q(Z|G_i, O)||P(Z|G_i, O)]. \tag{4}
$$

Here, since nodes in the original and consistent ego-graphs, i.e., $G_i$ and $\hat{G}_i$, can be different, we parameterize the encoder $Q(Z|\cdot)$ as a GNN that maps all the nodes in the whole graph $G$, i.e., $G_i \cup G_i^a$, to a latent Gaussian matrix $Z \in \mathbb{R}^{N \times K_Z}$. Afterward, $P(G_i^c|\cdot)$ decodes the consistent ego-graph $\hat{G}_i$ from $Z$ by first calculating the inner product between the latent variable of node $i$, i.e., $Z_i$, and the latent variables of the rest of the nodes in the whole graph, i.e., $Z$, and then sampling the one-hop neighbors for $\hat{G}_i$ from the Bernoulli distribution, i.e., $\mathbf{a}_i^T \sim Bernoulli(sigmoid(Z_i^T \cdot Z))$. Gumbel-softmax trick (Jang et al., 2017) is introduced to reduce the sampling variance. We then update the adjacency matrix of the whole graph $G$, i.e., $\mathbf{A}$, into $\mathbf{A}_{new}$ by setting $\mathbf{A}_{i:} := \mathbf{a}_i$. The $K$-hop consistent ego-graph $\hat{G}_i$ can be extracted from updated adjacency matrix $\mathbf{A}_{new}$.

### 3.2 Substitute Constraints and Objectives

However, since exemplar consistent ego-graphs $\hat{G}_i$ cannot be obtained due to local structural uncertainty, the expected log-likelihood term, i.e., $\mathbb{E}_{Q(Z|\cdot)}[P(\hat{G}_i|Z, O)]$, in Eq. (4) cannot be directly optimized to train the VEGAE-based SPN. Therefore, we resort to substitute objectives to replace the expected log-likelihood term in Eq. (4) to form a feasible objective to learn both $P(G_i^c|\cdot)$ and $Q(Z|\cdot)$. Specifically, we propose the following three substitute objectives:

- **Closeness.** Since the observed ego-graph $G_i$ contains certain information (although incomplete) on the interactions between unit $i$ and other units, the generated candidate ego-graph $G_i^c$ should preserve these information useful for inferring the true interference mechanism.

- **Consistency.** Assume that we can identify a set of stable nodes $\mathcal{N}_s$ in $G$ whose ego-graphs are probably consistent with the interference mechanism, the generated candidate ego-graph $G_i^c$ should follow the distribution of ego-graphs that are consistent with the interference mechanism.

- **Optimality.** Since we are looking for bounds of spill-over effects for the nodes under local structural uncertainty, $G_i^c$ should maximize/minimize the individual spill-over effect while satisfying the closeness and consistency criteria.

In this scenario, the above three criteria can be imposed on the candidate ego-graphs $G_i^c$ generated from the VEGAE-based SPN, i.e., $\mathbb{E}_{Q(\cdot|G_i,\cdot)}[P(G_i^c|\cdot)]$, where $P$ and $Q$ can be optimized s.t. the generated $G_i^c$ satisfy the three criteria, instead of directly maximizing the infeasible expected loglikelihood term $\mathbb{E}_{Q(\cdot|G_i,\cdot)}[P(G_i^c|\cdot)]$ with $G_i^c = \hat{G}_i$ as with the first term in Eq. (4).

Formulating the **closeness** constraint is straightforward, as we can define a differentiable function $Dis$ that measures the distance between the generated candidate ego-graph $G_i^c$ and the observed ego-graph $G_i$, where the expectation $\mathbb{E}_{\mathbb{E}_Q[P]}[(Dis(G_i^c, G_i))]$ can be used to measure the closeness. In this paper, $Dis$ is implemented as the mean square error (MSE) between the adjacency matrices (in the node space of the whole graph $G$) of $G_i^c$ and $G_i$.

### 3.3 Consistency Discriminator and Adversarial Objective

To achieve the **consistent** constraint, we introduce a discriminator $D : G_i^c \to \{0, 1\}$ that judges whether a given local structure around an arbitrary node $i$, i.e., $G_i^c$, is consistent with the interference mechanism. Since exemplar consistent ego-graphs $\hat{G}_i$ cannot be obtained for $i \in \mathcal{N}_u$, we assume that a set of stable nodes $\mathcal{N}_s$ can be identified in $G$, whose local structure is consistent with the interference mechanism. We denote the empirical distribution of the consistent ego-graphs of the stable nodes as $P_N(G_j^N) = \frac{1}{|\mathcal{N}_s|}, \forall j \in \mathcal{N}_s$. Here, we note that the identification of stable node sets is an untestable assumption. However, depending on the specific type of graph $G$, we can rely on many testable implications of consistency, e.g., the time that the node joins the graph, the node activity, and the node degree, etc. to determine the node set $\mathcal{N}_s$. We empirically find that $\mathcal{N}_s$ does not need to be perfectly accurate: it suffices as long as it exhibits less uncertainty than the nodes in $\mathcal{N}_u$.

The discriminator $D$ is implemented as a GNN. Considering that node sets could be different for different generated candidate ego-graphs $G_i^c$, we first embed $G_i^c$ into the adjacency matrix $\mathbf{A}$ of the whole graph $G$ as the input of $D$ (to ensure gradients can be backpropagated to the VEGAE-based SPN). We then conduct graph convolution only for the target node $i$, which outputs a score that denotes the consistency of the ego-graph $G_i^c$. $D$ is trained to assign the correct labels to the consistent ego-graphs for nodes in $\mathcal{N}_s$, i.e., $G_j^N \sim P_N(G_j^N)$, and the VEGAE-generated ego-graphs $G_i^c \sim \mathbb{E}_{Q(Z|G_i,O)}[P(G_i^c|Z,O)]$, which is implemented as the following adversarial objective:

$$\mathcal{L}_{adv}((P,Q), D) = \mathbb{E}_{P_N(G_j^N)}[\ln D(G_j^N)] + \mathbb{E}_{\mathbb{E}_{Q(Z|G_i,O)}[P(G_i^c|Z,O)]}[\ln(1 - D(G_i^c))]. \tag{5}$$

In addition, we can use the supremum of Eq. (5) w.r.t. the discriminator network $D$, i.e., $m = \max_D \mathcal{L}_{adv}((P,Q), D)$ to measure the discrepancy between the distribution of ego-graphs generated by VEGAE, i.e., $\mathbb{E}_{Q(Z|G_i,O)}[P(G_i^c|Z,O)]$ and the empirical consistent ego-graph distribution $P_N(G_j^N)$. When the VEGAE-generated ego-graphs for uncertain nodes follow the same distribution as the consistent ego-graphs, the theoretical minimum of $m$ can be achieved (Goodfellow et al., 2020). Therefore, we introduce $\max_D \mathcal{L}_{adv}((P,Q), D)$ to serve as the consistency constraint that, combined with the closeness constraint, determines the search space of probably consistent ego-graphs for nodes in $\mathcal{N}_u$.

### 3.4 Learning to Bound the Spill-over Effects

With the closeness and consistency criteria introduced above, we are ready to introduce the final criterion, **optimality**, which learns to bound the individual spill-over effect for nodes $i \in \mathcal{N}_u$ with local structural uncertainty. Specifically, the criterion can be formulated as maximizing/minimizing a spill-over effect estimator $\hat{\tau}^s(G_i^c, X_i, X_{-i}, T_i, T_{-i})$ that estimates the individual spill-over effect for node $i$ based on the VEGAE-generated ego-graphs $G_i^c$. Here, to ensure the accuracy of the estimator $\hat{\tau}^s$, we further constrain it to be *faithful* to the observed outcomes $Y$ on the graph $G$.

The connection between the spill-over estimator $\hat{\tau}^s$ and the faithfulness criterion is not evident. Fortunately, if the generated candidate ego-graph $G_i^c$ is consistent, which can be promoted with the consistency-promoting constraints, we can use the response function $\phi$ to connect faithfulness with $\hat{\tau}^s$. Specifically, if we define the response estimator as $\hat{\phi}(G_i^c, X_i, X_{-i}, T_i, T_{-i})$, which can be used to estimate the observed outcomes $Y_i$, the spill-over estimator $\hat{\tau}^s$ can be represented with $\hat{\phi}$ as:

$$\hat{\tau}^s = \hat{\phi}(G_i^c, X_i, X_{-i}, T_i, T_{-i}) - \hat{\phi}(G_i^c, X_i, X_{-i}, T_i, \mathbf{0}). \tag{6}$$

Given the above analysis and derivations, the lower bound for the individual spill-over effect of an uncertain node $i \in \mathcal{N}_u$ that simultaneously satisfies the closeness, consistency, optimality and faithfulness criteria introduced above can be formally formulated as follows:

$$\begin{aligned} \min_{P,Q} \ &\mathbb{E}_{G_i^c \sim \mathbb{E}_{Q(Z|G_i, \cdot)}[P(G_i^c|Z, \cdot)]} \Big[ \hat{\tau}^s(G_i^c, X_i, X_{-i}, T_i, T_{-i}) \Big], \\ s.t. \ &(i) \ \mathbb{E}_{\mathbb{E}_{Q[P]}}[Dis(G_i^c, G_i)] \leq \epsilon_d, \\ &(ii) \ \max_D \mathcal{L}_{adv}((P, Q), D) \leq m + \epsilon_c, \\ &(iii) \ \hat{\phi} = \min_{\hat{\phi}} \sum_j \mathcal{L}_{obs}(\hat{\phi}(G_j^c, X_j, X_{-j}, T_j, T_{-j}), Y_j), \end{aligned} \tag{7}$$

where the KL term in Eq. (4) is omitted for simplicity. Specifically, in Eq. (7), $\epsilon_d$ controls the maximum tolerance of the expected drifts between the VEGAE-generated ego-graph $G_i^c$ and the observed ego-graph $G_i$, whereas $\epsilon_c$ controls the tolerance of $G_i^c$ drifting away from the empirical consistent ego-graph distribution $P_N$ specified by the stable nodes.

Eq. (7) represents a complex bi-level optimization objective, which can be intuitively understood as follows: For all the ego-graphs $G_i^c$ generated by the VEGAE-based SPN that are at most $\epsilon_d$ away from the original ego-graph $G_i$, and $\epsilon_c$ away from the consistent graph distribution of the stable nodes, what is the minimum (maximum if $\min_{P,Q}$ is flipped as $\max_{P,Q}$) induced spill-over effect $\tau_s$ that we can get if the spill-over estimator $\hat{\tau}_s$ is faithful to the observed outcomes on the network (faithfulness achieved with loss $\mathcal{L}_{obs}$ of $\hat{\phi}$ minimized on the observed outcomes $Y$ as *(iii)*).

### 3.5 EM-based Objective

Directly optimizing Eq. (7) is difficult, as it involves complex interactions and nested constraints among the VEGAE-based ego-graph generative networks $P/Q$, the discriminator network $D$, and the spill-over effect estimation network $\hat{\phi}$. Therefore, we propose an expectation maximization (EM)-based solution to alternately optimize $P$, $Q$, $D$, and $\hat{\phi}$ in an iterative manner, where the training process can be substantially stabilized. Specifically, in the **e-step**, we replace the $K$-hop ego-graph of node $i \in \mathcal{N}_u$ in the adjacency matrix of $G$, i.e., $\mathbf{A}$, with the current probably consistent ego-graph generated by VEGAE, i.e., $G_i^c$, and optimize the response function estimator $\hat{\phi}$ as follows:

$$\textbf{e-step:} \min_{\hat{\phi}} \sum_j \mathcal{L}_{obs}(\hat{\phi}(G_j^c, X_j, X_{-j}, T_j, T_{-j})), Y_j) + \eta \cdot WD(P(Z^{\hat{\phi}}|T = 0), P(Z^{\hat{\phi}}|T = 1)). \tag{8}$$

Specifically, we introduce the Wasserstein distance penalty to balance the node representations $Z^{\hat{\phi}}$ learned in the response function estimator $\hat{\phi}$ between the nodes in the treatment group and non-treatment group

Table 1: Statistics of the datasets in the main paper.

|           | # Nodes | # Links   | % of treated |
|-----------|---------|-----------|--------------|
| Simulated | 2,000   | 20,000    | 49.77        |
| Flickr    | 7,575   | 236,582   | 47.41        |
| Company   | 70,965  | 3,847,963 | 79.82        |

(Shalit et al., 2017) (the calculation of $Z^{\hat{\phi}}$ will be introduced in the next sub-section). Eq. (8) is a tractable surrogate for the expectation-based objective in Eq. (7), using reparameterized samples from the generator via Gumbel-softmax trick introduced in Section 3.1. Using more samples would reduce approximation variance at the cost of increased computation. Afterward, in the **m-step**, we fix the response estimator $\hat{\phi}$ as $\bar{\phi}$ (and therefore the spill-over estimator $\hat{\tau}^s$ as $\bar{\tau}^s$) and optimize the generator and discriminator networks $P$, $Q$, and $D$ via the following partial objective:

$$\textbf{m-step:} \min_{P,Q} \max_{\lambda \geq 0} \max_{\gamma \geq 0} \max_{D} \mathbb{E}_{\mathbb{E}_Q[P]} \Big[ \underbrace{\bar{\tau}^s(G_i^c, X_i, X_{-i}, T_i, T_{-i})}_{\text{(3) min. the spill-over}}$$

$$+ \underbrace{\lambda \cdot \big( Dis(G_i^c, G_i) - \epsilon_d \big)}_{\text{(1) close to original graph}} \Big] \tag{9}$$

$$+ \gamma \cdot \Big( \underbrace{\mathcal{L}_{adv}((P,Q), D) - m - \epsilon_c}_{\text{(2) as if generated from consistency dist.}} \Big).$$

In practice, we simplify two of the nested optimizations, i.e., $\max_{\lambda \geq 0}$, $\max_{\gamma \geq 0}$, in Eq. (8) by introducing $\lambda$ and $\gamma$ as hyperparameters and removing the original hard constraints $\epsilon_d$ and $\epsilon_c$ to further improve the training stability. Generally, larger values of $\lambda$ and $\gamma$ correspond to smaller values of $\epsilon_c$ and $\epsilon_d$ (i.e., lower tolerance of violation of closeness and consistency criteria), where the influence of both $\lambda$ and $\gamma$ will be thoroughly discussed in the empirical study.

### 3.6  Response Function Estimator

The response function estimator $\hat{\phi}$ is composed of two modules, *(i)* confounder representation learning and *(ii)* potential outcome prediction. In the confounder representation learning module, we introduce a multi-layer perceptron (MLP), i.e., $MLP(X_i)$ and a GNN, i.e., $GNN(X_{-i}, T_{-i}, G_i^c)$ to capture the latent confounders from both the individual perspective (i.e., based on $X_i$ only) and the spill-over perspective (i.e., based on $X_{-i}$, $T_{-i}$, and $G_i^c$). Here, we note that $Z_i^{\hat{\phi}} = [MLP(X_i), GNN(X_{-i}, T_{-i}, G_i^c)]$ is the representation balanced in the **e-step** objective defined in Eq. (8) with the $WD$ term. Afterward, in the potential outcome estimation module, we introduce two more MLPs, i.e., $f_{t_i}$, for $t_i \in \{0, 1\}$, to estimate the potential outcome that corresponds to $T_i = t_i$. The final estimator $\hat{\phi}$ can be formulated via $MLP$, $GNN$, and $f_{t_i}$ as follows:

$$\hat{\phi}(G_i^c, X_i, X_{-i}, T_i, T_{-i}) = f_{T_i} \left( [MLP(X_i), GNN(X_{-i}, T_{-i}, G_i^c)] \right). \tag{10}$$

Here, we note that Eq. (10) is flexible enough to consider the non-linear interactions between the individual direct effect and the spill-over effect (instead of a linear additive model of both effects). Accordingly, based on Eqs. (6) and (10), the spill-over effect estimator $\hat{\tau}^s$ can be formulated as:

$$\hat{\tau}^s = f_{T_i} \left( [MLP(X_i), GNN(X_{-i}, T_{-i}, G_i^c)] \right) - f_{T_i} \left( [MLP(X_i), GNN(X_{-i}, \mathbf{0}, G_i^c)] \right). \tag{11}$$

Finally, stitching together Eqs. (4), (9), (10), and (11) introduced above provides the final optimization objective of the proposed learning to bound spill-over effect framework.

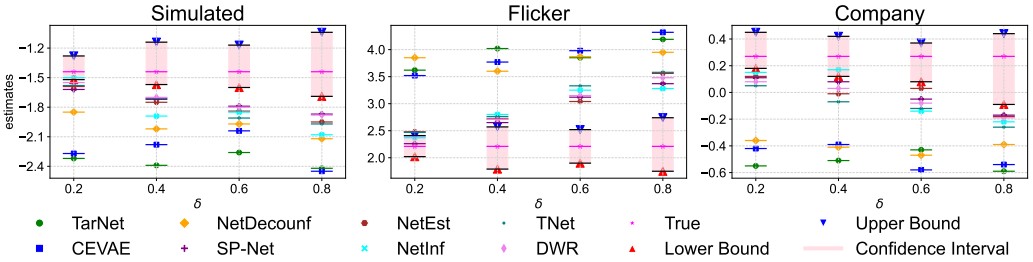

Figure 1: Comparison with baselines under different local levels of local structural uncertainty.

## 4 Empirical Study

### 4.1 Datasets

In this paper, we include three datasets to verify the effectiveness of the proposed framework: *(i)* a simulated dataset, *(ii)* a semi-simulated Flickr dataset (Ma et al., 2021), and *(iii)* a semi-simulated real-world dataset collected from the Company to estimate the influence of job Ads campaign in job marketplace[3]. Note that ground-truth spill-over effects are unavailable in real-world data because counterfactual outcomes are unobservable. Following standard practice in causal inference, we therefore use simulated and semi-simulated settings for quantitative evaluation and use the Company dataset to study practical applicability under realistic graph complexity. We note that the difficulty level of the three datasets gradually increases, as *(i)* considers a proof-of-concept linear additive response model, *(ii)* introduces non-linear interactions between individual and spill-over effects, and *(iii)* preserves the real-world complexity of interactions and considers multi-hop spill-over of treatment effects.

**Covariates, Graph, and Outcomes.** For the simulated dataset, we first generate the covariates as $\mathbf{X} \sim \mathcal{N}(\mathbf{0}, \mathbf{I}_{K_X})$, where for the Flickr dataset and the Company dataset, we use LDA (Blei et al., 2003) to reduce the dimension of the node textual features to $\mathbb{R}^{K_X}$ (where we set $K_X = 10$). We generate the adjacency matrix $\mathbf{A}$ for the simulated dataset by selecting max $N_A$ terms in $\mathbf{XX}^T$ to ensure an average degree of ten. For *Flickr* and *Company* datasets, real-world user connections are recorded as the adjacency matrix $\mathbf{A}$. The treatment $t_i$ for a node $i$ is simulated as follows:

$$t_i \sim Bernoulli\left(sigmoid\left(\mathbf{w}_t \cdot [\mathbf{X}_i + \lambda \cdot \bar{\mathbf{X}}_{ngh}]\right)\right), \tag{12}$$

where $\bar{\mathbf{X}}_{ngh}$ is the average covariates of the neighboring nodes and $\lambda$ is set to 0.1. For the Company dataset, user $i$ is treated if job postings from a certain company have been shown to the user in the previous week. For the *simulated* dataset, we consider a proof-of-concept simple linear additive response function to simulate the potential outcomes under networked interference:

$$Y_i^{T_i, T_{-i}} = T_i \phi^d\left(X_i\right) + \kappa \times \phi^s(X_{-i}, T_{-i}, G_i) + \epsilon_{Y_i}. \tag{13}$$

In Eq. (13), $\phi^d\left(X_i\right) = \mathbf{w}_d X_i$ is the individual response and the spill-over response $\phi^s$ is defined as:

$$\phi^s(X_{-i}, T_{-i}, G_i) = \frac{1}{|\mathcal{N}_i|} \sum_{j \in \mathcal{N}_i} T_j \times \phi^d\left(X_i\right), \tag{14}$$

where $\mathbf{w}_{\{d,s\}} \in \mathcal{N}(\mathbf{0}, \mathbf{I}_{K_X})$, and $\mathcal{N}_i$ denotes the neighbor set of node $i$. For the *Flickr* dataset, we add non-linearity and interference between $\phi^d$ and $\phi^s$ in the simulated response function as follows:

$$\begin{aligned}
Y_i^{T_i, T_{-i}} = T_i \phi^d\left(X_i\right) + \phi^s(X_{-i}, T_{-i}, G_i) + \frac{\kappa}{2} \cdot \\
(\phi^s)^2(X_{-i}, T_{-i}, G_i) + \frac{\kappa}{2} \phi^d\left(X_i\right) \phi^s(X_{-i}, T_{-i}, G_i) + \epsilon.
\end{aligned} \tag{15}$$

---

[3]Company name omitted per double-blind review policy.

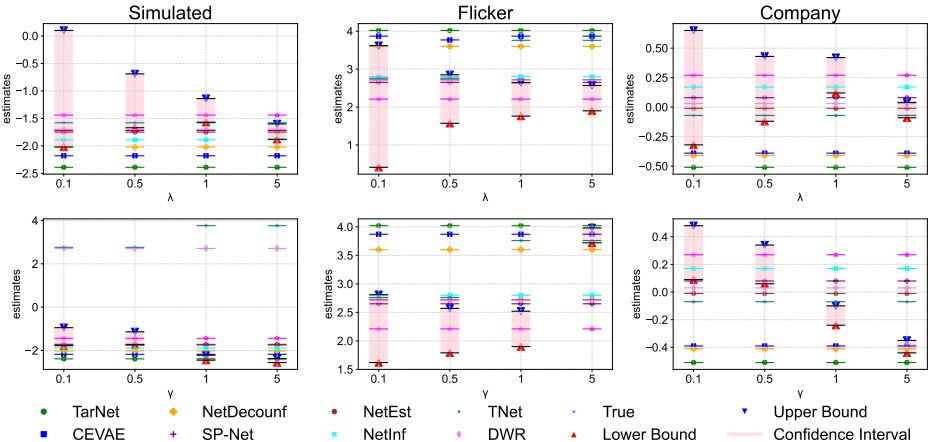

Figure 2: Comparison with baselines under different strength of Closeness and Consistency constrains.

Finally, for the *Company* dataset, we treat $\phi^d$ in Eq. (15) as an MLP and $\phi^s$ as a GNN and learn them by fitting on the observed data collected from the Company, such that the real-world information can be maximally preserved. The detailed statistics of the datasets are summarized in Table 1.

**Local Structural Uncertainty.** For the *simulated* and *Flickr* datasets, we randomly select $100 \times \theta\%$ noisy nodes as $\mathcal{N}_u$, where $100 \times \delta\%$ of their connections are randomly removed to simulate the local structural uncertainty. The influence of $\delta$ and $\theta$ (with the default of $\theta$ set as 2%) will be thoroughly discussed in subsections 4.2 and 4.4. For the *Company* dataset, we treat the cold-start members (joined within the late $100 \times \theta\%$ among all the users) as $\mathcal{N}_u$ and treat the predicted structure based on an internal social network completion model as the consistent ego-graphs $\hat{G}_i$. Please note that since the missing edges do not necessarily confine to the nodes in $\mathcal{N}_u$, the nodes in $\hat{\mathcal{N}}_s = \mathcal{N}/\mathcal{N}_u$ are not completely consistent. However, these nodes in $\hat{\mathcal{N}}_s$ contain substantially less uncertainty than nodes in $\mathcal{N}_u$, and we show that they can still effectively help estimate the bounds of spill-over effects.

## 4.2 Comparison with Baselines

**Baselines.** We include three classes of baselines for comparison. **(i)** The first class is traditional treatment effect estimators ignorant of the graph structure, e.g., TARNet (Shalit et al., 2017) and CEVAE (Louizos et al., 2017). **(ii)** The second class only considers graph structural information to estimate the latent confounders but is ignorant of the spill-over effect, e.g., NetDeconf (Guo et al., 2020). **(iii)** The third class uses GNN to model the spill-over effects and introduces adversarial learning or Hilbert-Schmidt Independence Criterion (HSIC) to balance the latent representation, e.g., NetEst (Jiang & Sun, 2022), NetItf (Ma & Tresp, 2021), SP-Net (Huang et al., 2023), TNet Chen et al. (2024), DWR Zhao et al. (2024). For all the methods, we report the average results of all uncertain nodes over five runs.

**Comparison and Analysis.** The comparison is illustrated in Fig. 1. From Fig. 1 we can find that, the upper bounds and lower bounds estimated by the proposed method cover the groundtruth spill-over effect values under all levels of local structural uncertainty. However, other baseline methods cannot provide accurate estimations and fall outside the bounds in most cases. Specifically, TARNet and CEVAE perform the worst as both methods completely ignore the graph structure. NetDeconf performs slightly better as it takes latent confounders in graph structure into consideration, but it still ignores the spill-over effect. NetEst and NetItf generally perform on par with each other. However, when local structural uncertainty is strong, the estimated spill-over effects still fall out of the bound.

## 4.3 The Strength of Closeness and Consistency Criteria as Constraints

In this part, we randomly sample the uncertainty level for the noisy nodes in the datasets from the uniform distribution $\delta \in Uniform(0.2, 0.8)$ and explore the influence of $\lambda$ and $\gamma$ that controls the strength of closeness

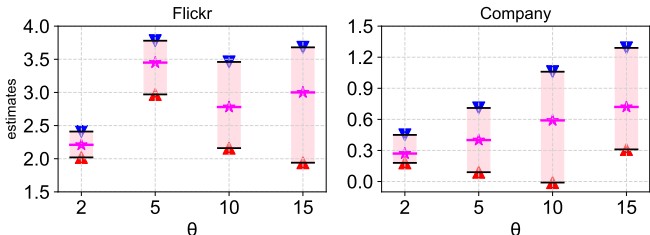

Figure 3: Bounds w.r.t. different % of uncertain nodes, where shaded interval shows the estimated bounds.

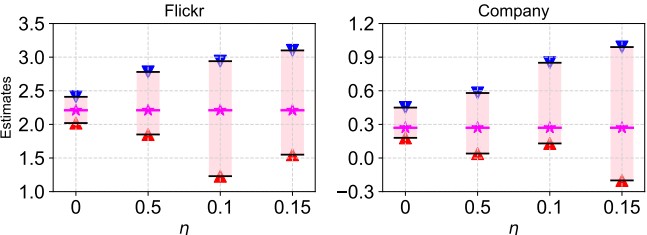

Figure 4: Bounds w.r.t. varied % of noise for stable nodes, where shaded interval shows estimated bounds.

and consistency to the bounds. The results are illustrated in Fig. 2. From Fig. 2. We can find that the tightness of the bound almost monotonically shrinks with larger values of $\lambda$ and $\gamma$ due to the smaller search space for the candidate probably consistent ego-graphs. However, the bias also tends to increase when $\lambda$ and $\gamma$ are overly large. The reason could be that, if the value of $\lambda$ becomes too large, the adversarial learning may be unstable and hinder the generator from learning useful information to generate the probably consistent ego-graphs. The analysis is similar for $\gamma$. When $\gamma$ is optimal, a suitable amount of information from the original local structure is incorporated to facilitate probably consistent ego-graph generation. However, when $\gamma$ grows too large, the generated ego-graphs become overly similar to the uncertain structure, which hinders the generation from being consistent with the true interference mechanism.

### 4.4 The Influence of Uncertain and Stable Nodes

In this section, we investigate the impact of the proportion of uncertain nodes in the graph (i.e., $\theta$) on our estimated bounds. As illustrated in Fig. 3, the results demonstrate that our proposed method effectively bounds the true spill-over effects and maintains robustness regardless of variations in the proportion of uncertain nodes. Furthermore, we examine scenarios where the consistency of the stable node set is compromised by randomly removing $\eta \times 100\%$ of the connections. The results, presented in Fig. 3 (with uncertain nodes fixed at noise level $\delta = 0.4$), reveal that even when the identified stable nodes contain noise, our method can still successfully bound the individual treatment effects for the uncertain nodes, as long as the local structure of the stable node set is more reliable than that of the uncertain nodes.

### 4.5 The Influence of Neighbors to the Bound

In this part, we explore the strength of neighbor influence on the performance of the proposed framework and various baselines, i.e., $\kappa$ in Eq. (13) and Eq. (15). The results are illustrated in Fig. 1 of the Appendix. From Fig. 1 we can find that the upper bounds and lower bounds estimated by the proposed method can still cover the ground truth spill-over effect values with varied strength of the influence from the neighbor nodes, despite that all the baselines fall out of the bound for most of the time. This further demonstrates the significantly enhanced robustness to local structural uncertainty of the proposed method.

## 5 Conclusions

In this paper, we propose learning to bound spill-over effects under local structural uncertainty. Specifically, we first introduce a variational ego-graph auto-encoder (VEGAE)-based structure proposal network to generate probably consistent ego-graphs for uncertain nodes. Accordingly, a spill-over effect estimator

is introduced to explore the upper and lower limits of the spill-over effect based on the ego-graph space defined by VEGAE, where generated ego-graphs are constrained with *closeness* and *consistency*, whereas the spill-over effect estimator is optimized based on *optimality* and *faithfulness*. Furthermore, an EM-based surrogate objective is proposed to improve training stability and efficiency.

## 6 Limitations

Our method studies spill-over effect estimation under local structural uncertainty, and we note several limitations that are worth further investigation in future work. First, the reported lower and upper bounds are optimization-based bounds over a learned feasible set of probably consistent ego-graphs, rather than statistical confidence intervals with formal coverage guarantees. This is because, under structural uncertainty, the true local interference structure is not directly observable, so the uncertainty in our setting comes not only from sampling variability, but also from ambiguity in the underlying ego-graph itself. Therefore we aim to learn lower and upper bounds over a feasible set of probably consistent ego-graphs, where the closeness and consistency criteria are introduced. Second, the framework relies on a stable-node set that provides relatively more reliable local structures for the consistency criterion. In practice, such nodes are identified using domain knowledge or observable proxies, and the quality of this approximation may affect the learned bounds. Finally, following prior spill-over effect estimation work, we assume that interference is localized within a $K$-hop ego-graph, where the choice of $K$ typically depends on application knowledge.

## 7 Broader Impact

This work is relevant to applications such as social, organizational, and economic networks, where spill-over effects may be important but the local interference structure is uncertain. In these settings, optimization-based lower and upper bounds can provide a more uncertainty-aware alternative to single point estimates. At the same time, the reliability of the resulting bounds depends on the quality of the graph information and modeling assumptions. We therefore view the proposed method as a tool for uncertainty-aware analysis that should be used together with domain knowledge, especially in high-stakes applications.

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

Table 2: Notation summary.

| Notation | Meaning |
|---|---|
| $G = (V, E)$ | Observed graph with node set $V$ and edge set $E$. |
| $G_i$ | Observed uncertain ego-graph centered at node $i$. |
| $\hat{G}_i$ | True but unobserved consistent ego-graph centered at node $i$. |
| $\hat{G}_i^c$ | Generated candidate probably consistent ego-graph for node $i$. |
| $G_{i,K}^a$ | Ambient graph associated with node $i$. |
| $\mathcal{N}_s$ | Stable-node set, whose observed ego-graphs are assumed to be relatively reliable. |
| $\mathcal{N}_u$ | Uncertain-node set, whose observed ego-graphs may be inconsistent. |
| $K$ | the $K$-hop ego-graph within which interference is assumed to occur. |
| $X_i$ | Covariates / features of node $i$. |
| $\mathbf{X}_{-i}$ | Covariates / features of the other nodes in the ego-graph of node $i$. |
| $T_i$ | Treatment assigned to node $i$. |
| $\mathbf{T}_{-i}$ | Treatments assigned to the other nodes in the ego-graph of node $i$. |
| $Y_i^{T_i, \mathbf{T}_{-i}}$ | Potential outcome of node $i$ under its own treatment $T_i$ and the treatment vector $\mathbf{T}_{-i}$ of the other nodes in its ego-graph. |
| $Y_i$ | Observed outcome of node $i$. |
| $\tau_i^d$ | Direct treatment effect for node $i$. |
| $\tau_i^s(t_i, \mathbf{t}_{-i})$ | Spill-over effect for node $i$ under target-node treatment $t_i$ and neighbor-treatment configuration $\mathbf{t}_{-i}$. |

# A   Proof of Theorem 2.1

In this section, we prove the identification of the individual direct and spill-over effects under Assumption 1, i.e., Unconfoundedness under Structural Recovery (S.R.), which implies *(i)* the ego-graph $\hat{G}_i$ is consistent with the interference mechanism for node $i$ (see Definition 1), and *(ii)* all confounders are contained either in node features $\mathbf{X}$ or node structures $\hat{G}_i$. The proof is as follows.

$$
\begin{aligned}
&\Phi(T_i, T_{-i}, \mathbf{x}_i, \mathbf{X}_{-i}, \hat{G}_i) \\
=&\mathbb{E}\left[Y_i^{T_i, T_{-i}}\Big| X_i = \mathbf{x}_i, X_{-i} = \mathbf{X}_{-i}, G_i = \hat{G}_i\right] \\
=&\mathbb{E}\left[Y_i^{T_i, T_{-i}}\Big| T_i, T_{-i}, X_i = \mathbf{x}_i, X_{-i} = \mathbf{X}_{-i}, G_i = \hat{G}_i\right] \\
=&\mathbb{E}\left[Y_i\Big| T_i, T_{-i}, X_i = \mathbf{x}_i, X_{-i} = \mathbf{X}_{-i}, G_i = \hat{G}_i\right].
\end{aligned} \tag{16}
$$

Specifically, the second step on the RHS of Eq. (18) is the direct application of Assumption 1, and the third step is based on the consistency rule of potential outcomes (Rubin, 1980). Based on Eq. (16), the individual direct effect of treatment $T$ on the outcome of node $i$ can be identified as follows:

$$
\begin{aligned}
\tau^d(\mathbf{t}_{-i}) &= \Phi(1, \mathbf{t}_{-i}, \mathbf{x}_i, \mathbf{X}_{-i}, \hat{G}_i) \\
&\quad - \Phi(0, \mathbf{t}_{-i}, \mathbf{x}_i, \mathbf{X}_{-i}, \hat{G}_i) \\
&= \mathbb{E}[Y|1, \mathbf{t}_{-i}, \mathbf{t}_i, \mathbf{X}_{-i}, \hat{G}_i] \\
&\quad - \mathbb{E}[Y|0, \mathbf{t}_{-i}, \mathbf{x}_i, \mathbf{X}_{-i}, \hat{G}_i].
\end{aligned} \tag{17}
$$

Similarly, the individual spill-over effect of node $i$ can be identified as follows:

$$
\begin{aligned}
\tau^s(t_i, \mathbf{t}_{-i}) &= \Phi(t_i, \mathbf{t}_{-i}, \mathbf{x}_i, \mathbf{X}_{-i}, \hat{G}_i) \\
&\quad - \Phi(t_i, \mathbf{0}, \mathbf{x}_i, \mathbf{X}_{-i}, \hat{G}_i) \\
&= \mathbb{E}[Y|t_i, \mathbf{t}_{-i}, \mathbf{x}_i, \mathbf{X}_{-i}, \hat{G}_i] \\
&\quad - \mathbb{E}[Y|t_i, \mathbf{0}, \mathbf{x}_i, \mathbf{X}_{-i}, \hat{G}_i],
\end{aligned} \tag{18}
$$

which finishes the proof of Theorem 2.1. Although for noisy nodes, Assumption 1 does not hold, Eq. (18) can still be used to connect the spill-over effect estimator on probably consistent ego-graphs and its faithfulness to the observed outcome, which makes the optimization of Eq. (7) feasible.

## B Proof of ELBO in Eq. (4)

In this section, we prove the ELBO of Eq. (4), i.e., the base optimization objective (if exemplar probably consistent ego-graphs $\hat{G}_i$ are available) that we improved upon, as follows.

$$
\begin{aligned}
\ln P(G_i^c|G_i,O) &\geq \mathbb{E}_Q[P(G_i^c|Z,O)] \\
&+ \mathbb{KL}[Q(Z|G_i,O)||P(Z|G_i,O)],
\end{aligned}
\tag{19}
$$

Proof.

$$
\begin{aligned}
&\ln P\left(G_i^c|G_i,O\right) \\
=&\ln \int_Z P\left(G_i^c, Z \mid G_i, O\right) dZ \\
=&\ln \int_Z Q\left(Z \mid G_i, O\right) \frac{P\left(G_i^c, Z \mid G_i, O\right)}{Q\left(Z \mid G_i, O\right)} dZ \\
\geq& \int_Z Q\left(Z \mid G_i, O\right) \ln \frac{P\left(G_i^c, Z \mid G_i, O\right)}{Q\left(Z \mid G_i, O\right)} dZ \\
=&\mathbb{E}_Q\left[\ln \frac{P\left(G_i^c, Z \mid G_i, O\right)}{Q\left(Z \mid G_i, O\right)}\right] \\
=&\mathbb{E}_Q\left[\ln \frac{P\left(G_i^c \mid Z, O\right) \cdot P\left(Z \mid G_i, O\right)}{Q\left(Z \mid G_i, O\right)}\right] \\
=&\mathbb{E}_Q\left[\ln P\left(G_i^c \mid Z, O\right)\right] - \mathbb{E}_Q\left[\ln \frac{Q\left(Z \mid G_i^c, O\right)}{P\left(Z \mid G_i^c, O\right)}\right] \\
=&\mathbb{E}_Q\left[\ln P\left(G_i^c \mid Z, O\right)\right] \\
&-\mathbb{KL}\left(Q\left(Z \mid G_i^c, O\right) \| P\left(Z \mid G_i^c, O\right)\right).
\end{aligned}
\tag{20}
$$

However, since exemplar probably consistent ego-graphs are not available for noisy nodes in $\mathcal{N}_s$, we introduce constraints (derived from measurable implications of consistency) on top of $G_i^c$ generated by VEGAE and combine them as the substitute objective.

