# OpenReview forum: "Bounding Spill-over Effect under Structural Uncertainty"
_TMLR — Rejected by TMLR_

### Review · Reviewer_J1wh · 2026-02-28

**Summary Of Contributions:**

The paper aims to provide valid lower/upper bounds for individual spill-over effects on graphs when the observed local neighborhood structure is uncertain or inconsistent with the true interference mechanism.
The paper formalizes causal inference with network interference using potential outcomes that depend on a node’s treatment and its neighbors’ treatments within a K-hop ego-graph, and they show identification of direct and spill-over effects under an Unconfoundedness under Structural Recovery assumption (i.e., the consistent ego-graph is observed and all confounders are captured by covariates and that structure).  When local structure is unreliable, they propose a Structure Proposal Network (SPN) parameterized as a variational ego-graph autoencoder that maps an observed (possibly incomplete) ego-graph to a distribution over “probably consistent” candidate ego-graphs (allowing different node sets). The paper constrains this candidate space with (i) closeness to the observed ego-graph (MSE between embedded adjacency matrices), and (ii) consistency via an adversarial discriminator trained to distinguish ego-graphs from a presumed stable-node set versus generated candidates. A response function estimator then predicts outcomes using both individual features and a GNN summary of neighbors, and the spill-over estimator is defined as the difference between the predicted outcomes under observed neighbor treatments and those under zeroed neighbor treatments. The paper then casts bound-finding as a constrained bilevel problem and proposes an EM-like alternating optimization with a representation-balancing penalty (Wasserstein distance) to stabilize learning.

**Audience:**

Yes

**Audience Explanation:**

The analysis is of interest to various communities within the TMLR audience, including network science researchers, researchers in causal and counterfactual methodology, and graph mining scientists.

**Broader Impact Concerns:**

I don't see direct concerns, but it would be good if the authors could discuss the sociological impact of their work.

**Claims And Evidence:**

Yes

**Claims Explanation:**

The paper explicitly targets cases where observed local graph structure is inconsistent with the true interference mechanism—something many spill-over estimators implicitly assume away.  The paper also argues that lower/upper bounds provide conservative/optimistic estimates, which is important in high-stakes settings where point estimates can be misleading under uncertainty. The empirical study varies the uncertainty level (edge removal), the proportion of uncertain nodes, and the noise in “stable” nodes, directly probing robustness claims rather than only reporting average-case performance. Across simulated, semi-simulated (Flickr), and a real “Company” dataset, the proposed method’s estimated lower and upper bounds consistently cover the ground-truth spill-over effects under varying levels of local structural uncertainty, while multiple baselines frequently miss. The paper also shows how constraint strengths affect bound tightness and bias.

**Requested Changes:**

The approach relies on identifying a set of “stable nodes” whose ego-graphs are consistent with the interference mechanism, but the paper acknowledges this is an untestable assumption and only gives heuristic criteria. If $N_s$ is misidentified, bounds may not be valid. Can you elaborate on this?

The identification is proven under Structural Recovery (consistent ego-graphs observed). The method seems to entail a learning-based constrained search. If so, what would the learned interval bounds be?

---

> ### Author Response · Authors · 2026-03-22
> **Response to Reviewer J1wh**
>
> We sincerely thank the reviewer for the careful reading and constructive comments. Below we provide itemized responses and explain how we will revise the manuscript accordingly.
>
> > **Comment 1.** On the identification of the stable-node set
>
> **Response:** Thank you for this important comment. We agree that the identification of the stable-node set is, in general, an untestable assumption. In our framework, the role of the stable-node set is to provide a source of relatively reliable local structures, whose ego-graph distribution is used to define the **consistency** criterion for uncertain nodes. Therefore, the key requirement is not that the stable-node set be perfectly known, but rather that it provide a more reliable structural reference than the uncertain-node set. Following your suggestion, we have clarified in the paper that the stable-node set could be approximated in practice using observable proxies of structural reliability, such as node age, node activity, node degree, or other domain-specific signals depending on the graph. We also make explicit that $\mathcal{N}_s\$ does not need to be perfectly accurate; in practice, it is sufficient that the nodes in $\mathcal{N}_s$ exhibit, on average, less structural uncertainty than those in $\mathcal{N}_u$.
>
> In addition, we now connect this discussion more explicitly to the experiment in Section 4.4, where we inject noise into the stable-node set. That result shows that the stable-node set remains helpful for learning the bounds as long as it is still less uncertain, on average, than the uncertain-node set.
>
> > **Comment 2.**  On the interpretation of the learned interval bounds
>
> **Response:** Thank you for this insightful comment. We would like to clarify that the learned lower and upper bounds in our framework are **optimization-based bounds over a learned feasible set of probably consistent ego-graphs**, rather than sharp identification regions or statistical confidence intervals with formal coverage guarantees. The ground-truth spill-over effect is always defined with respect to the **true but unknown consistent ego-graph**. However, under local structural uncertainty, this true ego-graph is not uniquely recoverable from observed data alone. Therefore, our goal is not to claim point identification under uncertainty, but to learn lower and upper bounds over a feasible set of **probably consistent** ego-graphs, where the proposed **closeness** and **consistency** criteria are introduced exactly for this purpose: closeness preserves the information already contained in the observed ego-graph, while consistency encourages candidate ego-graphs to align with the distribution of consistent ego-graphs from the stable nodes.
>
> > **Requested Changes**
>
> **Response:** According to the reviewer’s suggestions, we have revised the paper as follows:
>
> 1. **Expanded the discussion of the stable-node assumption.**
>    We now clarify more explicitly that the stable-node set is an untestable assumption, explain how it may be approximated in practice using observable proxies of structural reliability, and discuss more carefully what happens when this set is imperfectly identified.
>
> 2. **Clarified the interpretation of the learned interval bounds.**
>    We now state more explicitly that the reported lower and upper bounds are **optimization-based bounds over a learned feasible set of probably consistent ego-graphs**, rather than formal causal bounds with coverage guarantees. We also distinguish more clearly between identification under Structural Recovery and the constrained search procedure used under local structural uncertainty (see Abstract, last paragraph of introduction, and the limitation section).
>
> 3. **Added broader impact discussion.**
>    We have added discussion of the sociological implications of applying spill-over bounds under structural uncertainty, especially in socially and economically important graph settings where inaccurate estimate may affect downstream decisions.

---

### Review · Reviewer_KcPX · 2026-03-11

**Summary Of Contributions:**

This paper addresses the problem of estimating individual spill-over effects in causal effect estimation. This relaxes a key assumption in the conventional setting in Rubin Causal Model (RCM) which assumes the treatment received by a node will not affect the outcome of another node. In this setting, the outcome of a node can be influenced by treatments received by its neighbor nodes (e.g., on a social network) which is formally defined as the spill-over effect. The main focus of this paper is on estimating such effect under structural uncertainty.

This is achieved via representing the (uncertain) network connecting variables as a graph. Each node is associated with an ego-graph which contains all other nodes that can be reached within K hops. It is assumed that spill-over to a node can only happen within its ego-graph. The nodes are categorized into either certain nodes or uncertain nodes. These ego-graphs can be viewed as observations of the true interference mechanism among nodes.

For a certain node, its ego-graph is the true ego-graph of the (oracle) interference network. Under ignorability conditioned on true ego-graph, the spill-over effect reduces to a difference of conditional expectations of observed outcomes and can be estimated via standard ML techniques (see Theorem 2.1). This is similar to the standard identification result in RCM where treatment (T) and outcome (Y) are assumed to be conditional independent given the node X.

For an uncertain node, its ego-graph can miss edges or contain spurious edges and Theorem 2.1 no longer applies. This is where the main contribution of this paper enters. The idea is to learn a generator that generates candidate ego-graphs for uncertain nodes. It is trained to match the distribution over ego-graphs of the certain nodes under the constraint that the generated ego-graphs remain sufficiently close to the observed ego-graph in the network (see closeness constraint). Essentially, this view treats the ego-graphs at uncertain nodes as random quantities whose distributions are unknown. This makes the resulting spill-over estimate is itself a random variable under the generator's distribution, whose uncertainty must be quantified.

To achieve this, the authors introduce another learnable estimator that predicts the outcome conditioned on the generated ego-graph. In principle, if the generator converges towards generating true ego-graph, the estimator should aim to estimate the outcome under the standard identification result in Theorem 2.1. Its parameters should be learned to fit the observed effect in the training data. However, the generator is also being learned so it is essentially uncertain on its own. As such, the authors propose to let it behave adversarially such that its parameters are updated to either minimizing or maximizing the estimated spill-over effect while freezing the generator.

Running such adversarial process twice will provide lower and upper bounds for the individual spill-over effect of each uncertain node. The entire system is trained via an EM-style alternation: the E-step fixes the generator and retrains the estimator on the current candidate graphs, while the M-step fixes the estimator and jointly updates the generator and discriminator to push the spill-over toward its extremes.

**Audience:**

Yes

**Audience Explanation:**

This paper addresses an interesting and practical problem in causal effect estimation. I believe it'd be of interest to researchers in causal learning. Overall, this paper has an interesting algorithmic perspective but at the same time, it seems to lack a rigorous theoretical grounding as I pointed out in my review above.

**Claims And Evidence:**

No

**Claims Explanation:**

Overall, this paper is interesting but also relatively hard to read. I think the authors might want to rethink the notation system to make the technical details more digestible.

The central idea on bounding the spill-over effect is quite creative and well-motivated. The experiments spanning three datasets with increasing complexity are reasonable. The empirical finding (Figure 4) that the bounds remain (empirically) valid even when the certain node set itself contains noise is practically important and somewhat reassuring (though it remains unclear why)

On the other hand, I have a few concerns:

1. The bound seems to lack formal guarantee. The paper presents the output interval [lower, upper] as bounds on the spill-over effect, but these are algorithmic bounds (outputs of an optimization procedure), not statistical bounds with any formal coverage guarantee. There is no theorem stating that the true spill-over lies within the reported interval with any specified probability. The validity of the bounds depends entirely on whether the closeness and consistency constraints adequately characterize the set of truly consistent ego-graphs, which is neither guaranteed nor formally analyzed. Perhaps this should discussed more.

2. The identification of the certain nodes appears ad-hoc (i.e., somewhat specific to the experiment datasets) and it's unclear how one would obtain this information in a generic situation. This heuristic identification of certain/uncertain nodes was discussed only briefly prior to Section 4.2. It'd be better if the authors could discuss more generalizable methods to distinguish certain/uncertain nodes.

3. Different candidate ego-graphs imply different neighbor sets, which means the counterfactual "set all neighbor treatments to zero" is defined over different sets of nodes for different candidates. This means the estimation is not invariant to structure uncertainty. This fundamental issue needs to be discussed.

4. From Eq. 7 to Eq. 8, it seems the expectation in Eq. 7 is replaced with a single sample estimate via Eq. 8. This seems to be a practical design choice but how this approximation affects the quality of the computed bounds is not discussed.

5. In addition to a lot of algorithmic approximations, this work also has made deeper assumptions on the interference structure. For example, it is assumed that interference only happens within K-hop graphs. But, how do we verify this? Another implicit assumption is that the observed graph is the interference but in practice, that might not be true. For example, social network edge might represent "friendship" while the actual interference channel operates through a different mechanism (i.e., whether the two "nodes" talk to each other about the treatment).

**Requested Changes:**

In addition to the above requests, I think the manuscript can be strengthened moreN/ with rigorous discussions on the following questions:

1. Can you provide any formal guarantee (even asymptotic) that the true spill-over effect lies within the reported bounds?
2. How sensitive is the method to the choice of K (the locality horizon)?
3. Will using multiple samples from the generator per E-step rather than a single point estimate improve the performance?
4. Can you provide more articulation on why do we assume the graph topology of uncertain nodes is similar to that of certain nodes? It seems plausible that in practice, the graph topology of uncertain nodes might be legitimately different from that of certain nodes.

Please also give some thoughts regarding simplifying the notation systems to improve the readability of the manuscript.

---

> ### Author Response · Authors · 2026-03-22
> **Response to Reviewer KcPX - Part 1**
>
> We sincerely thank the reviewer for the careful reading and constructive comments. Below we provide itemized responses and explain how we will revise the manuscript accordingly.
>
> > **Comment 1.** On the lack of formal guarantee for the reported bounds
>
> **Response:** Thank you for this insightful comment. We acknowledge that the lower and upper bounds of spill-over effects studied in this paper are **optimization-based bounds over a learned feasible set of probably consistent ego-graphs**, rather than sharp identification regions or statistical confidence intervals with formal coverage guarantees. The reason is that, under local structural uncertainty, the true interference neighborhood for uncertain nodes is generally **not uniquely recoverable from observed data alone**. As a result, the uncertainty here is not only sampling uncertainty around a fixed estimand, but also structural uncertainty about which ego-graph defines the estimand itself. For this reason, our method does not aim to claim statistical coverage. Instead, we define a tractable feasible graph space through the **closeness** and **consistency** criteria, and then explores the lower and upper limits of the spill-over effect over that space.
>
> We also acknowledge that the resulting bounds depend on these criteria. However, since the true distribution of consistent ego-graphs is inherently unavailable under structural uncertainty, some tractable and heuristic assumptions are unavoidable. In our framework, **closeness** is introduced to preserve the useful information already contained in the observed uncertain ego-graph, while **consistency** encourages the generated ego-graphs to follow the distribution of the relatively reliable ego-graphs. Following your suggestion, we have made this distinction clearer in both abstract and introduction and further discuss the difference between our optimization-based bounds and classical statistical bounds in the limitations section.
>
> > **Comment 2.** On the identification of certain / stable nodes
>
> **Response:** Thank you for raising this important point. We agree that the identification of stable versus uncertain nodes is application-dependent and untestable. In our framework, the stable-node set serves as a source of relatively consistent local structures, whose ego-graph distribution is used to define the **consistency** criterion for uncertain nodes. Therefore, the key requirement is not that the stable-node set be perfectly known, but rather that it provide a more consistent local structural information with the true interferece mechanism than the uncertain-node set.
>
> Following your suggestion, we now emphasize more clearly in the methodology that the identification of the stable-node set is an assumption of the framework. In addition, depending on the type of graph, we can rely on observable proxies that provide useful evidence of relative structural reliability, such as the time a node joins the graph, node activity, node degree, or other domain-specific signals, to approximate the set $\mathcal{N}_s$​. We further clarify that $\mathcal{N}_s$​​ does not need to be perfectly identified: in practice, it is sufficient that the nodes in $\mathcal{N}_s$​​ exhibit, on average, less structural uncertainty than those in $\mathcal{N}_u$​. In addition, we now connect this discussion more explicitly to the experiment in Section 4.4, where we inject noise into the stable-node set. That result shows that the stable-node set remains helpful for learning the bounds as long as it is still less uncertain than the uncertain-node set.
>
> > **Comment 3.** On different candidate ego-graphs implying different neighbor sets
>
> **Response:** Thank you for this insightful observation. We would like to clarify that the ground-truth spill-over effect in our framework is always defined on the **true but unknown consistent ego-graph**. The issue is not that the estimand itself changes, but that under local structural uncertainty, the true ego-graph is unavailable, and different candidate ego-graphs could all be probably consistent, which may induce different neighbor sets when approximating it. Our method therefore searches over a feasible set of **probably consistent** ego-graphs to approximate this unknown true local structure. The resulting lower and upper bounds are defined over this feasible candidate space, but they are all intended to approximate the same underlying spill-over effect defined by the true interference structure.

---

> ### Author Response · Authors · 2026-03-22
> **Response to Reviewer KcPX - Part 2**
>
> > **Comment 4.**  On the approximation from Eq. (7) to Eq. (8)
>
> **Response:** Thank you for this helpful comment. We acknowledge that Eq. (8) is a practical approximation to the expectation-based objective in Eq. (7), introduced to make optimization tractable and stable. In particular, we approximate the expectation using reparameterized samples from the ego-graph generator via the Gumbel-Softmax trick (Jang et al., 2017), which is a standard low-variance strategy in variational inference for Bernoulli-type variables such as graph edges. This approximation mainly affects the variance of optimization, rather than changing the underlying objective itself.
>
> In principle, using more samples can further reduce the variance of the approximation, but it also increases the computational cost. Since we already observe good empirical coverage of the ground-truth spill-over effect with one-sample optimization, we adopt this choice in our experiments as a practical trade-off between stability and efficiency. Following your suggestion, we have clarified this approximation more carefully in the revised manuscript.
>
> > **Comment 5.** On the K-hop interference assumption
>
> **Response:** Thank you for this constructive feedback. The assumption that interference is localized within a $K$-hop neighborhood is standard in prior spill-over effect estimation work, including the baselines considered in our paper, namely **TARNet, CEVAE, NetDeconf, SP-Net, NetEst, NetInf, TNet, and DWR**. In our experiments, we follow the same $K$-hop setting as these baselines so that all methods are evaluated under a consistent interference horizon. This assumption is also natural in many applications, since if two units are too far apart to interact through the graph, it is less plausible that treatment on one unit can directly influence the outcome of the other.
>
> We also acknowledge that the choice of $K$ generally relies on the experimenter’s domain knowledge, since it is difficult to verify directly from observational data. In practice, $K$ should be chosen to reflect the range within which interference is believed to exist. Following your suggestion, we have clarified this point more explicitly in the revised manuscript.
>
> > **Comment 6.** On the assumption that the observed graph is the interference graph
>
> **Response:** Thank you for highlighting this important limitation. We would like to clarify that this corresponds to a different level of uncertainty from the one studied in our paper. Our focus is on **local structural uncertainty**, namely the case where the observed graph is a meaningful proxy for the interference structure but may contain missing or spurious local connections. We do **not** aim to address the more fundamental setting where the wrong graph itself is chosen for modeling spill-over effects, i.e., where the observed graph is not the relevant interference graph at all.
>
> As in prior work on graph-based spill-over estimation, we assume that the observed graph is selected based on domain knowledge and is informative about the underlying interference mechanism. From an experimental design perspective, choosing an appropriate graph for the spill-over process is therefore a prerequisite, rather than the source of uncertainty that our method is intended to handle.
>
> > **Comment 7.** On why the topology of uncertain nodes is assumed to be similar to that of certain nodes
>
> **Response:** Thank you for this helpful comment. We would like to clarify that our framework does **not** assume that uncertain nodes and stable nodes have the same marginal distribution of topology. Instead, the consistency criterion is applied **conditional on node features, neighboring-node features, and the observed local graph structure**. The stable-node set therefore serves as a relatively reliable reference distribution under comparable local information, rather than imposing unconditional topological similarity. This is consistent with our assumption of **strong ignorability under structural recovery**: after conditioning on the recovered local structure and relevant covariates, treatment can be viewed as randomized (i.e., the distributional discrepancy between the nodes vanishes). We have clarified this point more explicitly in the revised manuscript.

---

> ### Author Response · Authors · 2026-03-22
> **Response to Reviewer KcPX - Part 3**
>
> > **Comment 8.** On notation and readability
>
> **Response:** Thank you for this helpful suggestion. We have carefully revised the notation throughout the paper and added a notation table in the Appendix according to your advice. In particular, we apologize that we cannot substantially simplify the notation for the potential outcomes $Y^{T\_i,\mathbf{T}\_{-i}}$ and the spill-over estimators $\tau^s(t_i,\mathbf{t}_{-i})$, since these expressions need to explicitly distinguish the treatment of the target node from the treatments of the other nodes in the ego-graph. This distinction is fundamental to the causal inference setting considered in the paper.
>
> That said, we have simplified the notation related to ego-graphs to improve readability. Specifically, we revise the observed uncertain ego-graph centered at node $i$ from $G^{sub}\_{i,K}$ to $G_i$ when there is no ambiguity in $K$, the consistent ego-graph from $G^{cln}\_{i,K}$ to $\hat{G}_i$, the probably consistent candidate ego-graph generated from VEGAE from $G^{cad}\_{i,K}$ to $\hat{G}^{c}\_i$, and the ambient ego-graph from $G^{amb}\_{i,K}$ to $G^a\_{i,K}$. We also revise the notation for the generated probably consistent candidate ego-graph to make it more visually distinct from the true consistent ego-graph. Overall, we believe these changes make the symbol system cleaner and improve the readability of the paper.
>
> > **Requested Changes**
>
> **Response:** According to the reviewer’s suggestions, we have revised the paper as follows:
>
> 1. **Clarified the interpretation of the reported bounds.**
>    We now explicitly state that the reported lower and upper bounds are **optimization-based bounds over a learned feasible set of probably consistent ego-graphs**, rather than statistical confidence intervals with coverage guarantees. We further explain that, under local structural uncertainty, the true interference neighborhood is generally not uniquely recoverable from observed data alone, so the uncertainty is not only sampling uncertainty around a fixed estimand, but also structural uncertainty about which ego-graph defines the estimand itself.
>
> 2. **Clarified the choice of the locality horizon $K$.**
>    We now state more explicitly that the $K$-hop interference assumption is a standard modeling assumption in prior spill-over effect estimation work, including the baselines in our paper. We also clarify that the choice of  $K$generally relies on domain knowledge, since it is difficult to verify directly from observational data, and that the framework should be interpreted conditional on the chosen locality horizon.
>
> 3. **Clarified the approximation from Eq. (7) to Eq. (8).**
>    We now explain that Eq. (8) is a practical approximation to the expectation-based objective in Eq. (7), implemented using reparameterized samples from the ego-graph generator via the Gumbel-Softmax trick. We also clarify that using more samples could further reduce the variance of optimization but would increase computational complexity, and that we adopt one-sample optimization in practice because it already provides good empirical coverage of the ground-truth spill-over effect.
>
> 4. **Strengthened the discussion of the similarity between stable and uncertain nodes.**
>    We now clarify that the framework does **not** assume that uncertain nodes and stable nodes have the same marginal topology. Instead, the consistency criterion is imposed **conditional on node features, neighboring-node features, and the observed local graph structure**, so that the stable-node set serves as a relatively reliable reference distribution under comparable local information.

---

### Review · Reviewer_Gmxj · 2026-03-11

**Summary Of Contributions:**

This paper studies spill-over effect estimation under local graph uncertainty. Instead of producing a single point estimate, it proposes to estimate lower and upper bounds on individual spill-over effects by generating candidate ego-graphs for uncertain nodes and optimizing over this feasible graph space. The problem is meaningful and practically relevant, and the bounded-estimation perspective is well motivated for uncertain graph settings. The method is also reasonably structured, and the simulation results are encouraging.

However, the paper has several weaknesses: the theoretical guarantees are limited in the main uncertain-structure setting; a key conceptual step is insufficiently justified, namely why the VEGAE decoder output should be interpreted as a valid candidate ego-graph rather than merely a generated graph; and the real-world evaluation is less conclusive because no ground-truth spill-over effect is available.

**Audience:**

Yes

**Audience Explanation:**

The problem is relevant to researchers in causal inference, graph machine learning, and uncertainty-aware estimation. The paper addresses an important limitation of existing interference-aware methods, namely their reliance on trustworthy graph structure. Even if the current evidence is not fully complete, the problem formulation and overall direction are likely to interest part of the TMLR audience.

**Broader Impact Concerns:**

I do not see major ethical concerns, but a brief broader impact discussion would be helpful. Since the motivating applications include social or economic graphs, the paper should mention possible risks of using inaccurate spill-over bounds in downstream decision-making.

**Claims And Evidence:**

No

**Claims Explanation:**

The simulation and semi-simulation results provide reasonable evidence that the method is more robust than point-estimation baselines when local graph structure is noisy. However, the theoretical justification is incomplete, since the paper proves identification only in the ideal consistent-structure case, not for the main uncertain-structure setting. The real-world experiment is also less convincing because no ground-truth spill-over effect is available.

**Requested Changes:**

1. Clarify whether the proposed outputs are formal causal bounds or empirically motivated uncertainty intervals.
2. Strengthen the discussion/theory for the uncertain-structure case, not only the ideal consistent case.
3. Be more careful in interpreting the real-world experiment, where no ground truth is available.

---

> ### Author Response · Authors · 2026-03-22
> **Response to Reviewer Gmxj - Part 1**
>
> We sincerely thank the reviewer for the careful reading and constructive comments. Below we provide itemized responses and explain how we will revise the manuscript accordingly.
>
> > **Comment 1.** On the theoretical guarantee in the uncertain-structure setting
>
> **Response:**  Thank you for this insightful comment. We would like to clarify that Section 2.2, _Ideal Case with Consistent Local Structure_, is only a preliminary theoretical analysis that shows prior work on spill-over effect estimation is identifiable only when the relevant interference structure to be known, or correctly recovered. It aims to reveal their weakness to show the motivation of our method.
>
> In our setting, however, the key challenge is exactly that for uncertain nodes, the observed ego-graph may be inconsistent with the true interference mechanism. In our case, the true interference neighborhood is generally **not uniquely recoverable from observed data alone**. Accordingly, our goal is not to claim point identification under uncertainty, but to learn lower and upper bounds over a feasible set of **probably consistent** ego-graphs, where the proposed **closeness** and **consistency** criteria are introduced exactly for this purpose: closeness preserves the information already contained in the observed ego-graph, while consistency encourages candidate ego-graphs to align with the distribution of consistent ego-graphs from the stable nodes.
>
>
> > **Comment 2.** On why the VEGAE decoder output can be treated as a candidate ego-graph
>
> **Response:** Thank you for this insightful comment. We would like to clarify that the output of a plain decoder alone would not justify treating a decoded ego-graph as a valid candidate interference structure. In our framework, the generator only serves to propose candidate ego-graphs. To make the generated ego-graph consistent, we start with Eq. (4) from the standard but ideal supervised setting, where the true consistent ego-graph is available as supervision. However, this setting is infeasible under local structural uncertainty, because the true consistent ego-graph is exactly what we do not observe. For this reason, we introduce surrogate criteria to determine whether a proposed ego-graph can be treated as a probably consistent candidate as supervision signal. Specifically, a decoded ego-graph is accepted only if it satisfies the constraints in Eq. (7), so that it lies in a feasible graph space under structural uncertainty. In particular, constraint **(i) closeness** preserves the information already contained in the observed ego-graph, while constraint **(ii) consistency** requires the generated ego-graph to align with the distribution of consistent ego-graphs from the stable nodes. Therefore, our method does not equate “decoded” with “valid.” Rather, it uses constrained generation to explore a tractable feasible space of probably consistent ego-graphs, over which the upper and lower bounds of spill-over effects are defined.
>
> > **Comment 3.**  On the real-world experiment without ground-truth spill-over effects
>
> **Response:** Thank you for this constructive feedback. In causal inference, it is common that real-world data do not provide ground-truth causal effects, since the counterfactual outcomes are inherently unobservable. For this reason, existing work (including TARNet, CEVAE, NetDeconf, SP-Net, NetEst, NetInf, TNet, and DWR, which are introduced in our paper as baselines)  typically relies on **simulated** or **semi-simulated** settings for quantitative evaluation, while using real-world data to assess practical applicability. Our empirical study follows the same practice: the paper includes a fully simulated dataset and a semi-simulated Flickr dataset for evaluation against simulated ground truth, and further includes the Company dataset to study a more realistic setting with richer real-world complexity and multi-hop spill-over effects. Accordingly, the role of the Company experiment is not to provide direct verification against a known true spill-over effect, but to examine whether the proposed framework remains applicable in realistic observational settings where  more complex multi-hop interference.
>
> To make this positioning clearer, according to your advice, we have revised the manuscript and the limitation section to explicitly note that ground-truth spill-over effects are not available in real-world data. Following standard practice in causal inference, we therefore evaluate the method using datasets with progressively increased complexity, from simulated to semi-simulated settings where the true spill-over effect is available, and then to the Company dataset, where the spill-over effect is simulated on top of a real graph to better reflect realistic multi-hop interference patterns.

---

> ### Author Response · Authors · 2026-03-22
> **Response to Reviewer Gmxj - Part 2**
>
> >  **Requested Changes**
>
> **Response:** Thank you for this constructive feedback. In response to the your insightful suggestions, we have revised the paper as follows:
>
> 1.  **Clarified the interpretation of the proposed outputs.**
>     We now explicitly state that the reported lower and upper bounds are **optimization-based bounds over a learned feasible set of probably consistent ego-graphs**, rather than sharp identification regions or statistical confidence intervals (see abstract, the first sentence in the last paragraph of introduction, and the limitation section).
>
> 2.  **Strengthened the discussion of the uncertain-structure setting.**
>     We now clarify that Section 2.2 and Theorem 2.1 serve only as a preliminary result for the ideal structural-recovery case to reveal the shortcoming of existing method, while the main focus of the paper is the uncertain-structure setting where the true interference structure is not uniquely recoverable from observed data (see the first paragraph of Section 2.2.1).
>
> 3.  **Revised the interpretation of the real-world experiment.**
> 	We now state more carefully that ground-truth spill-over effects are not available in real-world data due to infeasible counterfactuals, and following standard practice in causal inference, the paper uses simulated and semi-simulated settings for evaluation against ground truth, and uses the Company dataset as a more realistic case study with richer multi-hop interference patterns rather than as direct validation against a known true effect (see both Section 4.1 and Limitations).

---

### Decision · Action_Editor_T1TN · 2026-04-16

**Recommendation:** Reject

**Additional Comments:**

The primary concern raised by reviewers is the lack of theoretical guarantees for the proposed bounds. To strengthen the paper, the authors may consider incorporating a formal definition of the structural uncertainty that the method is designed to address. In the current formulation, the feasible set of “probably consistent” ego-graphs is defined implicitly through the generator and surrogate constraints, which makes the resulting bounds appear more algorithm-induced rather than grounded in a clearly specified theoretical framework. Providing a more explicit formulation of the uncertainty, such as through a probabilistic model over ego-graphs $P(\hat{G}_i|G_i)$, could help clarify the problem setting and better justify the interpretation of the resulting bounds.

**Audience:**

Yes

**Audience Explanation:**

All reviewers agree that the problem is relevant and that the paper would be of interest to at least part of the TMLR audience, particularly researchers in causal inference, graph machine learning, and uncertainty-aware estimation. Even those with concerns about the technical justification note that the problem formulation and overall direction are interesting and meaningful to the community.

**Claims And Evidence:**

No

**Claims Explanation:**

The reviewers generally agree that this submission addresses a relevant and important problem. The proposed methodological idea is novel and well-motivated, and the empirical results are generally strong. However, there are consistent concerns regarding the theoretical justification of the method, which remain unresolved after the author response. In particular, two reviewers highlight the lack of theoretical guarantees and lean toward rejection. The third reviewer leans toward acceptance but does not directly address this concern in their review.

**Resubmission Of Major Revision:**

The authors may consider submitting a major revision at a later time.